# The assessment of fundus image quality labeling reliability among graders with different backgrounds

**Kornélia Lenke Laurik-Feuerstein**[1]☯, **Rishav Sapahia**[2]☯, **Delia Cabrera DeBuc**[2]*, **Gábor Márk Somfai**[3,4,5]*

**1** Department of Ophthalmology, Klinikum Ernst von Bergmann, Potsdam, Germany, **2** Miller School of Medicine, Bascom Palmer Eye Institute, University of Miami, Miami, Florida, United States of America, **3** Department of Ophthalmology, Stadtspital Zürich, Zurich, Switzerland, **4** Spross Research Institute, Zurich, Switzerland, **5** Department of Ophthalmology, Semmelweis University, Budapest, Hungary

☯ These authors contributed equally to this work.
* somfaigm@yahoo.com (GMS); dcabrera2@med.miami.edu (DCD)

**Data Availability Statement:** Data will only be available after acceptance.

**Funding:** This study was supported in part by the Finker Frenkel Legacy Foundation (DCD), a NIH

## Abstract

### Purpose

For the training of machine learning (ML) algorithms, correctly labeled ground truth data are inevitable. In this pilot study, we assessed the performance of graders with different backgrounds in the labeling of retinal fundus image quality.

### Methods

Color fundus photographs were labeled using a Python-based tool using four image categories: excellent (E), good (G), adequate (A) and insufficient for grading (I). We enrolled 8 subjects (4 with and 4 without medical background, groups M and NM, respectively) to whom a tutorial was presented on image quality requirements. We randomly selected 200 images from a pool of 18,145 expert-labeled images (50/E, 50/G, 50/A, 50/I). The performance of the grading was timed and the agreement was assessed. An additional grading round was performed with 14 labels for a more objective analysis.

### Results

The median time (interquartile range) for the labeling task with 4 categories was 987.8 sec (418.6) for all graders and 872.9 sec (621.0) vs. 1019.8 sec (479.5) in the M vs. NM groups, respectively. Cohen's weighted kappa showed moderate agreement (0.564) when using four categories that increased to substantial (0.637) when using only three by merging the E and G groups. By the use of 14 labels, the weighted kappa values were 0.594 and 0.667 when assigning four or three categories, respectively.

### Conclusion

Image grading with a Python-based tool seems to be a simple yet possibly efficient solution for the labeling of fundus images according to image quality that does not necessarily

Center Grant No. P30-EY014801 to the University of Miami (DCD), and by an unrestricted grant to the University of Miami from Research to Prevent Blindness, Inc. (DCD). No additional external funding was received for this study.

**Competing interests:** The authors have declared that no competing interests exist.

require medical background. Such grading can be subject to variability but could still effectively serve the robust identification of images with insufficient quality. This emphasizes the opportunity for the democratization of ML-applications among persons with both medical and non-medical background. However, simplicity of the grading system is key to successful categorization.

## Introduction

Medical imaging gains an ever-increasing role in modern medicine urging the need for human expertise for image interpretation and data evaluation [1–3]. In many clinical specialties there is a relative shortage of such expertise to provide timely diagnosis and referral. Machine Learning (ML) has emerged as an important tool for healthcare due to the rapid evolution of artificial intelligence (AI) in various medical fields, such as ophthalmology and radiology. The importance of high-quality data is pivotal when it comes to training new AI/ML models, using datasets that are mindfully designed and annotated. Moreover, such datasets should report on sociodemographic features of the patients, on the in- and exclusion criteria, the labelling process and the labelers themselves [4, 5].

Globally, sight-loss costs $3B annually, with an incidence that is projected to triple by 2050, 80% of which is preventable [6]. That is, severe vision loss is needlessly experienced by too many patients; regarding diabetic retinopathy (DR) alone, appropriate treatment can reduce the risk of blindness or moderate vision loss by more than 90% [7]. The compounding limitation of availability of retina specialists and trained human graders is also a major problem worldwide. Consequently, given the current population growth trends it is inevitable that automated applications with limited human interaction will expand over time [3].

Machine Learning has already provided multiple clinically relevant applications in ophthalmology which include image segmentation, automated diagnosis, disease prediction, and disease prognosis [8–10]. Ophthalmology is particularly suitable for ML because of the crucial role of imaging, where fundus photographs, optical coherence tomography, anterior segment photographs, and corneal topography can be applied to conditions such as diabetic retinopathy, age-related macular degeneration, glaucoma, papilledema, and cataracts [3, 8–10].

In order to provide sufficient data for the training and validation of ML algorithms, correctly labeled ground truth data are inevitable. In an effort to model the democratization of retinal image labeling, in this pilot study we assessed the performance of graders with different backgrounds in the labeling of retinal fundus image quality using a custom-built grading tool constructed in Python. We also assessed the ability to identify images with poor quality and obtained feedback from the graders to assess their impressions.

## Materials and methods

Color fundus photographs were chosen from a total of 18,145 color fundus images intended for the future training of an AI based algorithm for image quality grading. Three datasets were used as a source for color fundus images: the dataset by Tao et al. [11], the EyePacs dataset [12] and finally a dataset of 984 color fundus images obtained from the Bascom Palmer Eye Institute. The images were taken with a variety of fundus cameras.

The study was performed in adherence to the guidelines of the Declaration of Helsinki, ethics approval was obtained for the study from the local ethics committee of the University of Miami. The images were collected with the participants' consent and were de-identified as

required by local regulatory requirements (e.g., the Health Insurance Portability and Accountability Act).

An image classification system was developed based on the previous work of Zapata et al., Fleming et al., Gulshan et al. and on the EyePACS image quality grading system [12–15]. The image grading criteria and their definitions can be seen in detail in Table 1. In short, four criteria were used to define image quality: focus, image field definition, brightness and artefacts. Based on the fulfillment of the criteria definitions, four groups were created: excellent (E, all criteria sufficiently fulfilled), good (G, maximum 2 criteria not fulfilled), adequate (A, 3–4 criteria not fulfilled but the retina is recognizable) and insufficient for grading (I, no retinal tissue visible on more than 50% of the image; no third generation branches of the retinal vasculature can be identified one-disc diameter to the fovea and optic nerve head) (S1 File). We randomly selected 200 images from a pool of 18,145 expert-labeled images with 50 images from each group. The images were previously labeled for image quality by two experts and served as the reference for the analysis using the standards described above. These experts were a board-certified ophthalmologist (KLLF) with experience in retinal imaging and a board-certified senior retina specialist (GMS) with retinal image grading experience in the telemedical screening of diabetic retinopathy. In case of any disagreement, GMS's decision served as a reference [16]. A representative image for each image quality group, along with its labels can be seen on Fig 1.

A tkinter based graphical user interface (GUI) tool was developed in Python by taking inspiration from the multiple open-source tools available. (https://docs.python.org/3/library/tkinter.html, last accessed 11. July, 2021) The main intent was to develop a simple, lightweight, on-prem image annotation tool with a smooth learning curve for medical professionals. A screenshot of the GUI of the tool is shown on Fig 2. The tool also enabled the timing of the grading of each image shown in a resolution of 900x1000 pixels, without the option of zooming in.

For the purposes of this study, the images were evaluated by 8 volunteers, 4 with medical background (3 ophthalmologists and 1 optometrist, aged 26–45 years–group M) and 4 without medical background (2 computer scientists, 1 lawyer and 1 teacher, aged 26–60 years–group NM). All volunteers had at least intermediate computer skills working with a PC on a daily basis. Prior to grading, the participants received a tutorial consisting of a detailed oral explanation of the task, supported by a PDF document describing retinal anatomy, the grading system with sample images including examples of different artefacts and the description of the GUI (S1 File). All graders were encouraged to keep the tutorial open or as a printout in order to facilitate the grading task (S1 File). The images were presented in the same, randomly selected order for all volunteers.

**Table 1. Image quality grading criteria.** Image quality was determined based on four relevant categories representing the standard for good quality color fundus images. All the following categories are necessary in order to grade retinopathy lesions in fullest.

| Grading categories | Definition |
|---|---|
| **Focus** | Details are present up to a level allowing to grade smallest retinal alterations e.g. microaneurysm, intraretinal microvascular abnormalities. The small retinal vessels within one-disc diameter around the fovea are depicted sharply. |
| **Illumination** | The amount of source light incident on the retina is correct for the visualization of smallest retinopathy lesions. There are no washed-out or dark areas that interfere with detailed grading. |
| **Image field definition** | The primary image field includes the entire optic nerve head (ONH) and macula. There is at least one optic disc diameter retina nasally and temporally from the ONH and macula, respectively. |
| **Artefacts** | Artefacts in the image acquisition such as: dust spots, arc defects, fingerprints, camera reflexes or eyelash images regardless whether they hindered image grading |

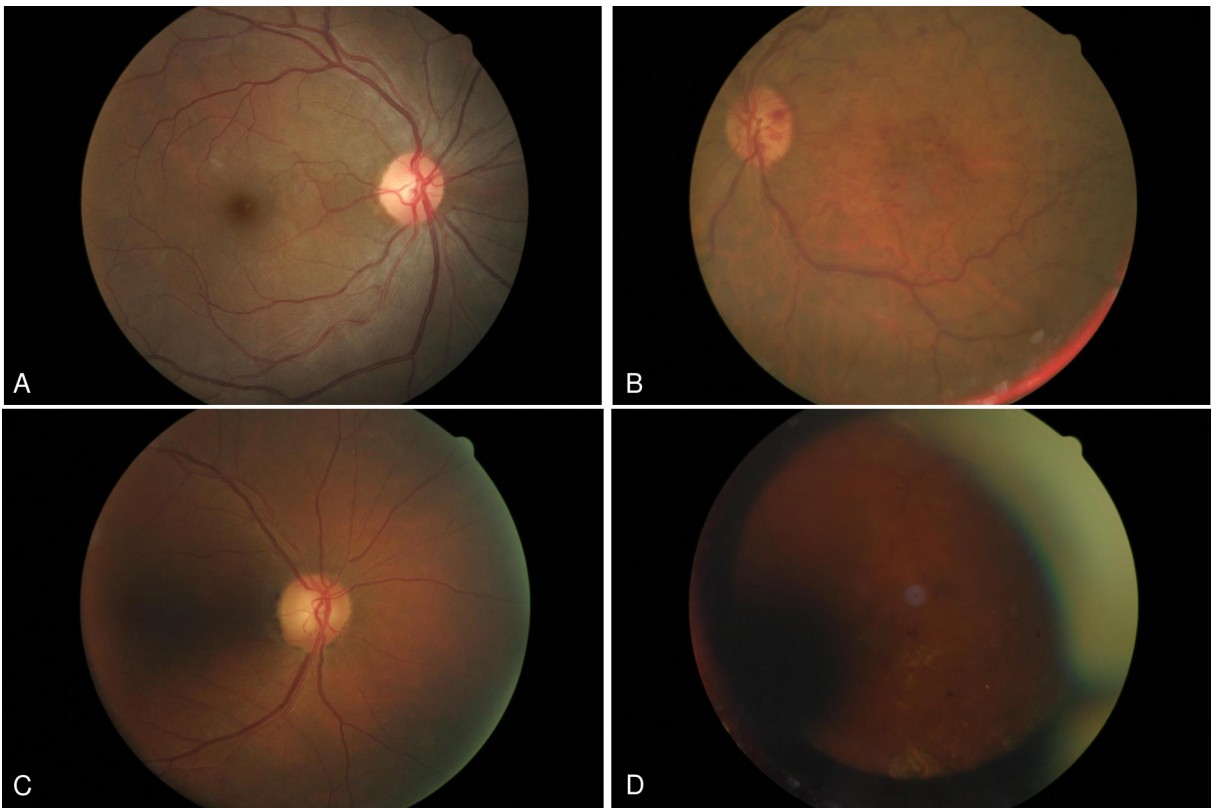

**Fig 1.** Representative color fundus images from the Excellent (A), Good (B), Adequate (C) and Insufficient for grading (D) categories. Image 1B fulfills criteria for "Good" due to the image field definition (decentered image) and peripheral artifacts; 1C qualifies as Adequate due to its poor illumination, off focus and due to insufficient image field definition (the image field does not contain enough of the retina temporal to the fovea). Image 1D was labelled as Insufficient as it neither depicts the optic nerve head nor makes it possible to visualize the third-generation vessel branches around the macula which, in turn, would not enable to detect retinal changes characteristic for diabetic retinopathy.

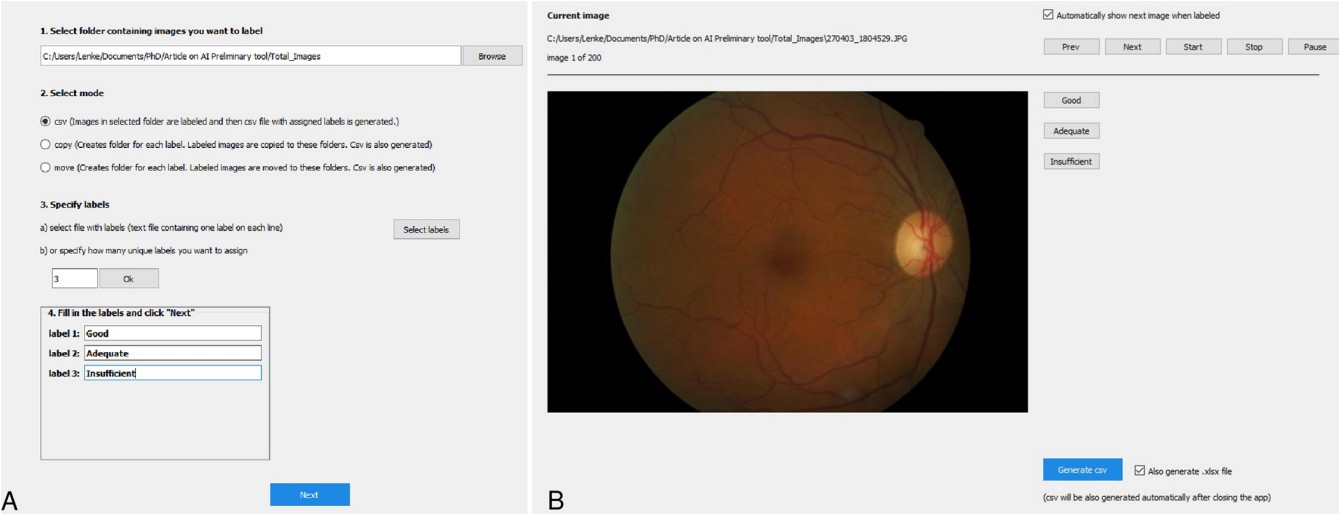

**Fig 2. Screenshots of the image labeling tool developed in Python demonstrating its function. A)** First, the folder containing the images is selected (1). In the next step the output of image labeling can be chosen (2) and finally the labels are specified (3). **B)** After setting the above parameters there are two options. Either navigate through with the "Prev" and "Next" buttons, or run a timed labeling round by selecting the option to "automatically show next image when labeled" and then the "Start" button. Upon closing the tool, a.csv and optionally.xls will be generated with the results.

**Table 2. Grading labels used for the objective grading round.** In order to decrease the inherent subjectivity of our study, our participants were asked to perform a second round of grading using predefined labels assigned to each category. In this round, the four categories were then complied similarly to the first round of grading.

| Grading categories | Labels used in the objective grading round |
|---|---|
| **Focus** | • Optimal<br>• Unsharp |
| **Illumination** | • Optimal<br>• Too dark<br>• Too light |
| **Image field definition** | • Optimal<br>• Missing Macula<br>• Missing Optic disc |
| **Artefacts** | • No artefacts<br>• Small pupil<br>• Dust spots<br>• Lash artefacts<br>• Camera artefacts<br>• Arc defects |

In order to assess image quality in a more objective way, the graders were also asked to perform a second round of grading with the tool using 14 labels (Table 2). Using these labels, the same groups were generated following the assessment (S1 File). After completion of the grading task the participants were asked to give a written feedback regarding their experiences.

The time needed for the grading was recorded for each volunteer. To compare the times needed for the labeling according to the four categories, we used the medians and interquartile range (IQR) due to the low number of graders in both groups.

For the analysis of the inter-rater agreement, Cohens's weighted kappa was calculated in the first grading round for the four categories (E/G/A/I) and in the second, objective round of grading using 14 labels. The 4 categories were compiled according to the categories of the previous grading system (E/G/A/I).

Cohens's weighted kappa was also determined with the excellent and good categories merged [(E+G)/A/I] for both grading rounds. To assess the agreement in selecting poor quality images both in the first and second round, Cohen's weighted kappa was calculated with merging groups E and G vs. A and I. The kappa values are presented as medians (IQR) for all the graders and for both groups (M, NM) in Table 3.

The statistical analysis was carried out with SPSS 28 (SPSS Inc, Chicago, IL).

## Results

The median time (IQR) for the labeling task was 987.8 sec (418.6) for all graders, and 872.9 sec (621.0) vs. 1019.8 sec (479.5) in the M vs. NM groups, respectively. The median time necessary

**Table 3. Inter-rater agreement in the two setups with different image quality category groups.** Cohens's weighted kappa was calculated for the grading using 4 image quality categories (Excellent (E)/ Good (G)/ Adequate (A)/ Insufficient (I)) and for the second round of grading using 14 labels. In the latter, the same 4 categories were compiled as in the first round of grading (E/G/A/I). Cohens's weighted kappa was also determined with E and G merged for both grading rounds [(E+G)/A/I]. To assess the agreement when distinguishing between poor quality images both in the first and second round, Cohen's weighted kappa was calculated with two merged groups (E and G vs. A and I). The kappa values are presented as medians (interquartile range) for all the graders and for both groups (medical, non-medical).

| | 4 Image quality grading criteria | | | 14 Predefined labels | | |
|---|---|---|---|---|---|---|
| | **4 groups** | **3 groups** | **2 groups** | **4 groups** | **3 groups** | **2 groups** |
| **Medical** | 0.590 (0.167) | 0.657 (0.116) | 0.715 (0.190) | 0.598 (0.053) | 0.669 (0.052) | 0.708 (0.126) |
| **Non-medical** | 0.554 (0.176) | 0.627 (0.147) | 0.625 (0.175) | 0.568 (0.085) | 0.612 (0.107) | 0.581 (0.127) |
| **Altogether** | 0.564 (0.163) | 0.637 (0.096) | 0.665 (0.178) | 0.594 (0.60) | 0.667 (0.80) | 0.670 (0.151) |

for the first 50 images (262.8 sec [125.7]) was somewhat longer compared to that required for the last 50 images (195.8 sec [108.3]). Graders with medical training finished labeling the last 50 images in median 28.6 seconds faster than the first 50 images. Graders without former medical training finished the last 50 images in median 38.25 seconds faster than the first 50 images.

The median time needed for the decision making per image (IQR) in the four categories was 3.35 sec (3.4), 3.8 sec (4.0), 4.0 sec (4.8) and 1.8 sec (2.0) for the E, G, A and I label, respectively. The median time for single decision was in all categories longer in the NM group (3.8 vs. 2.4 sec, 4.3 vs. 3.2 sec, 4.6 vs. 3.1 sec, 2.0 vs. 1.6 sec for the E, G, A and I in the NM vs. M groups respectively).

The Cohen´s weighted kappa showed moderate agreement among our graders when using 4 categories (0.564 for all graders and 0.590 vs. 0.554, for the groups M and NM, respectively). This increased to substantial with merging the E and G groups (0.637 for all graders, and 0.657 vs. 0.627 for the groups M and NM, respectively). Cohen's weighted kappa further increased with merging groups E and G vs. A and I (0.665 for all graders, and 0.715 vs. 0.625 for the groups M and NM, respectively) (Table 3).

In the second round of grading using 14 labels, the 4 categories were compiled according to the categories of the previous grading system (E/G/A/I). Here the agreement by Cohen's weighted kappa was moderate when using four categories (0.594 for all graders and 0.598 vs 0.568 for the groups M and NM, respectively), substantial when merging E and G groups (0.667 for all graders and 0.669 vs 0.612 for the groups M and NM, respectively), whereas merging the A and I groups only increased agreement minimally altogether and among medical graders (0.670 for all graders and 0.708 vs. 0.581, for the groups M and NM, respectively) (Table 3).

Assessment of the post-task reports showed that all graders found using the Python image labeling tool easy and the grading task not complicated to handle. However, 2 of our 8 graders without former experience in programming with Python needed additional support to launch the grading tool for the first time. These 2 graders experienced difficulties while installing the package manager used for our application and had difficulties using command lines for opening the application (as opposed to the more common practice of launching applications via clicking an icon). Therefore, all our participants were also provided with a detailed "program launching supplement" and oral explanation in the second round of grading that eliminated these difficulties. Five graders found the size and resolution of the images somewhat small for grading retinal lesions. Our participants found the grading task and the written (S1 File) and oral tutorials comprehensible. None of the graders reported decision fatigue during the grading task. Five graders reported to have felt a noticeable improvement in grading even with 200 images and found the experience motivating.

Our graders with medical background (three ophthalmologists and one optometrist) found a relatively high percentage of fundus images with an at least partially missing optic nerve head. In contrast, the role of the small artefacts without significant influence on the grading of potential retinopathy lesions was questionable. As for our image quality classification system, two graders (one from the M and one from the NM group) expressed their desire for more clarity on the definition of focus and illumination concerning the visibility of smaller retinal vasculature. Altogether, the feedback of our participants was positive regarding the labeling task and the tool itself.

## Discussion

Our paper presents the results of a pilot study implemented via a Python-based image quality grading tool. In this study, we used randomly chosen fundus images from our study dataset

that has already been labelled by two experts for image quality. We trained graders with and without ophthalmological background in a standardized fashion for the grading criteria and for the use of our image quality grading tool. Compared to current literature in the field, we did not only evaluate inter-rater repeatability but also analyzed the time spent on single decisions and the complex grading task by distinguishing image quality features with participants lacking medical background and undergoing only minimal training.

The grading task of 200 images took approximately 17 minutes, graders with an ophthalmological background did the task faster than those without. The time necessary for the first 50 images was somewhat longer than for the last 50 images. Besides the learning effect, we explain this with a better handling of the application that some of the participants also mentioned in their feedbacks, while the variable proportion of images from the different categories (requiring different decision times) could also be in the background. Decision fatigue or apathy leading to faster response with less consideration might also be a factor for this, however, is rather unlikely due to the short time (less than 30 minutes in the first and around 45 minutes in the second round) necessary for the entire grading. The participants required the least time for the grading of insufficient images, almost half of that needed for the other groups. In each of our 4 groups (E/G/A/I) the grading took somewhat longer for the participants without medical background.

Our results show that a high level of agreement [17] can be achieved already by a very short training period and smaller set of retinal photographs, even in the case of medically untrained people. The use of only three groups instead of four substantially increased the agreement of the grading, pointing towards the importance of simple grading categories in such a setting as it might be difficult to discern fine features in image quality that could be, in turn, of less importance when implementing robust deep learning techniques. In general, the participants were satisfied with the Python-based tool and expressed concerns that we are planning to incorporate in future developments of our grading software.

Numerous semi-automated, computerized analysis techniques were developed to reduce the costs and the workload of expert image graders [18] and deep learning algorithms have recently been and are currently being adopted for the detection of different retinal diseases [19]. The Bhaktapur Retina Study showed moderate to almost perfect agreement between mid-level ophthalmologic personnel (n = 2) and ophthalmologist in grading color fundus photographs for hemorrhages and maculopathy [20]. Similarly, a study of the Center for Eye Research Australia showed that non-expert graders (2-months intensive training, n = 2) are able to grade retinal photos with at least moderate level of DR with high accuracy [21]. McKenna et al. found good accuracy among non-medical graders (1-month intensive training for grading purposes, n = 4) and relatively poor performance by rural ophthalmologists in a DR grading study in China [22]. A novel tool for generating ground truth data is via crowdsourcing where volunteers can perform the analysis online either for free or for a fee-for-service. Brady et al. have several reports on using Mechanical Amazon Turk as an effective and inexpensive means to rapidly identify ocular diseases such as DR or follicular trachoma on photographs [23, 24].

In consensus with the results of Brady et al. [23], graders with minimal training can contribute to a cost-effective and rapid grading of retinal photographs. In our study, it took non-medical graders in median 17 minutes to grade the 200 images. We can estimate that the labeling of a dataset containing 20,000 retinal images with 20 graders without previous education in the field of ophthalmology would take approximately 90 minutes. However, one must not underestimate grading fatigue as a possible influencing factor, as mentioned above. We believe, however, this was not the case in our current study due to the relatively short processing time (less than 30 minutes for the original task) and due to the total number of 200 images

that would not reach the typical level of exhaustion associated with grader decision fatigue, as shown by Waite et al. [25]. Even better training results can be achieved with trained personnel; however, repeatability and accuracy (specificity and sensitivity) in this case should also be assessed along with images from different datasets.

It is important to underline the potential weaknesses of our study. We included only 8 volunteers for the assessment of the labeling task that could be higher in order to simulate a real-world setting of crowdsourcing. However, we did not intend to go towards crowd-sourcing, we rather aimed to provide an analysis of the usability of our labeling tool and grading system, instead. Also, the size of the cohort is comparable or even larger than that reported earlier [23]. The size of the dataset consisting of 200 fundus images may be considered small. However, other similar studies used the same order of data [21].

Another weakness could be the assessment of the optic nerve head and the fovea which may play a decisive role in various pathologies. However, our primary goal with the development of our grading tool was the assessment of general image quality and thus, the optic nerve head or fovea has been graded within the image field definition as an equally important image quality factor as focus, illumination and artefacts. Another important motivation of our study was to investigate whether there was a considerable disadvantage in outsourcing the grading task to people without medical background. In concordance with our purpose, we considered the presence of an artefact enough to meet the criteria regardless of their clinical relevance or effect on the image grading possibilities.

A particular strength of our study is that we used a pool of fundus images derived from different fundus cameras, representing a setting close to real life where the labeling of image quality should ideally be independent from the device used form imaging.

In the future, we are planning to expand our work and based on the post-grading comments of our participants we will adjust our Python tool to optimize image resolution. The image grading protocols and checklists will also need to be reevaluated with special attention on the role of the missing optic nerve head, the visibility of the small macular blood vessels and the relevance of the small artefacts irrelevant for grading. To minimize susceptibility to handling difficulties and standardize the grading process, we are currently developing a web-based annotation tool that will allow additional zooming and illumination options. This will enable us to implement the learnings of our study on a broader basis, in a simpler setting. Our further purpose is to use our data for the training of a convolutional neural network for image quality assessment and grading of DR and other retinal diseases.

With the democratization of AI, all its advantages and disadvantages will become available to a broad audience. The role of crowdsourcing increases in numerous projects that require harnessing human intelligence. This role, however, raises a new question in the context of health-care data: how safe is it to leave medically relevant decisions to "untrained" personnel? The significance of our work is the possible guidance on how to start producing meaningful ground truth data even with partially untrained participants for training and testing the new generations of deep learning algorithms. Our results show that it is sufficient to include fewer grading categories, that is, to work with a less detailed but more robust grading system, without the need of labeling single image quality criteria. Second, such a relatively simple tool can help to address the issue of poor-quality images, as all datasets, including the public repositories (such as EyePACS and MRL Eye) or the numerous smaller, institutional datasets, contain a certain amount of poor-quality images that are not gradable. Our approach could be an important asset not only in the grading of image quality but could also serve as a first screening step to filter out these images with poor quality. According to our results, this step (quality grading and filtering ungradable images) could be realized without expert level knowledge or longer training by simply employing "citizen scientists". We hope that our findings and our

application will help the training and development of robust artificial neural networks for the labeling of fundus image quality.

## Supporting information

**S1 File. Detailed grading tutorial used for the training of our graders.**
(PDF)

**S1 Dataset. Detailed annotation and time data used for the evaluation presented.**
(XLSX)

## Acknowledgments

This manuscript is dedicated to the loving memory of László Laurik who himself performed the grading task as part of the group of graders with no medical background. The authors also wish to thank all volunteers (Viviane Guignard, Rozina Ida Hajdú, Martin Lörtscher, Irén Szalai, Marco Feuerstein and László Miklós Laurik), for their time and effort performing the analyses. Meeting presentation: part of the results in this paper were presented as a poster presentation at the 114. Jahreskongress der Schweizerischen Ophthalmologischen Gesellschaft, August 25–27., 2021

## Author Contributions

**Conceptualization:** Kornélia Lenke Laurik-Feuerstein, Rishav Sapahia, Delia Cabrera DeBuc, Gábor Márk Somfai.

**Data curation:** Kornélia Lenke Laurik-Feuerstein, Rishav Sapahia, Gábor Márk Somfai.

**Formal analysis:** Kornélia Lenke Laurik-Feuerstein.

**Investigation:** Kornélia Lenke Laurik-Feuerstein.

**Methodology:** Kornélia Lenke Laurik-Feuerstein, Rishav Sapahia.

**Software:** Rishav Sapahia.

**Supervision:** Rishav Sapahia, Delia Cabrera DeBuc, Gábor Márk Somfai.

**Validation:** Gábor Márk Somfai.

**Writing – original draft:** Kornélia Lenke Laurik-Feuerstein, Delia Cabrera DeBuc, Gábor Márk Somfai.

**Writing – review & editing:** Kornélia Lenke Laurik-Feuerstein, Rishav Sapahia, Delia Cabrera DeBuc, Gábor Márk Somfai.

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
