## [Decision Letter · Decision Letter 0]

26 Dec 2021

PONE-D-21-35940The assessment of fundus image quality labeling reliability among graders with different backgroundsPLOS ONE

Dear Dr. Somfai,

Thank you for submitting your manuscript to PLOS ONE. After careful consideration, we feel that it has merit but does not fully meet PLOS ONE’s publication criteria as it currently stands. Therefore, we invite you to submit a revised version of the manuscript that addresses the points raised during the review process.

We look forward to receiving your revised manuscript.

Kind regards,

Andrzej Grzybowski

Academic Editor

PLOS ONE

Journal Requirements:

5. We note that Figures 1A, 1B, 1C 1D, 2A and 2B in your submission contain copyrighted images. All PLOS content is published under the Creative Commons Attribution License (CC BY 4.0), which means that the manuscript, images, and Supporting Information files will be freely available online, and any third party is permitted to access, download, copy, distribute, and use these materials in any way, even commercially, with proper attribution. For more information, see our copyright guidelines: http://journals.plos.org/plosone/s/licenses-and-copyright.

a. You may seek permission from the original copyright holder of Figures 1A, 1B, 1C, 1D, 2A and 2B to publish the content specifically under the CC BY 4.0 license. 

Reviewers' comments:

Reviewer's Responses to Questions

**Comments to the Author**

1. Is the manuscript technically sound, and do the data support the conclusions?

Reviewer #1: Partly

Reviewer #2: Yes

2. Has the statistical analysis been performed appropriately and rigorously? 

Reviewer #1: I Don't Know

Reviewer #2: No

3. Have the authors made all data underlying the findings in their manuscript fully available?

Reviewer #1: No

Reviewer #2: Yes

4. Is the manuscript presented in an intelligible fashion and written in standard English?

Reviewer #1: Yes

Reviewer #2: Yes

5. Review Comments to the Author

Reviewer #1: Overview

Thank you for giving me the opportunity to review the article. I hope the following suggestions will be helpful and will strengthen the work.

The study evaluates the image quality assessment of color fundus photographs among graders with medical and non-medical backgrounds. The topic is interesting and relevant considering the scarcity of labeled data for the development of machine learning models and the importance of imaging quality for the accuracy and applicability of the algorithms. The manuscript is generally well presented, but there are several issues which needs to be addressed:

Major comments

Page 4, line 51-60. Quotation marks should be used for direct quotes. Consider rewriting the first paragraph of the Introduction as the following sentences were written using the same words as the references:

- Page 4, line 51. “Medical imaging is expanding globally at an unprecedented rate leading to an ever-expanding quantity of data that requires human expertise and judgement to interpret and triage”. De Fauw et al (2018).

- Page 4, line 55. “Benchmark datasets are essential for computational research in healthcare. These datasets should be created by intentional design that is mindful of social and health system priorities. If a deliberate and systematic approach is not followed, not only will the considerable benefits of clinical algorithms fail to be realized, but the potential harms may be regressively incurred across existing gradients of social inequity.” (T. Panch et al. 2020)

The study was conducted “in an effort to model the democratization of retinal image labeling” (page 5, line 78). To do that, the authors use a phyton grading tool that requires some degree of programming knowledge (page 8, line 162), which could make it difficult for people without specific computing skills to participate in the image labeling process.

The imaging grading criteria used in the study take into consideration anatomical structures (e.g., optic nerve head, fovea and macula), and imaging artefacts (e.g., eyelashes and fingerprints) whose identification require a clinical background or specific training and a certain degree of experience. According to the authors, “prior to grading the participants received a tutorial consisting of oral explanation of the task, supported by a pdf document describing the grading system with sample images and the description of the GUI.” (Page 6, line 120). In addition to the tutorial on image quality requirements, has any training been carried out to enable participants to recognize retinal structures and artefacts? If yes, detailed information on how the training was carried out should be provided (e.g., hours of training and grading guidelines). If not, the interpretation of the grading criteria by participants without medical background would be affected, as well as the results of the study.

Page 7, line 130. Was the agreement calculated between study participants or between each participant and the ground truth (i.e., expert’ labels)? Was the weighted kappa applied to assess agreement when grading image quality (an ordinal outcome)?

Page 9, line 186. “The time necessary for the first 50 images was higher than for the last 50 images which shows the learning curve of the grading task.” Based on the data presented, it is not possible to conclude that faster grading is a consequence of the learning curve as there are other possible explanations for this result (e.g., grading fatigue).

Minor comments

Page 2, line 28. “The performance of the grading was timed and the accuracy was assessed. Accuracy was also assessed with the excellent and good categories merged.” Since kappa is not a measure of accuracy, it is not possible to say that the accuracy was assessed.

Page 2, line 32 and page 7, line 140. Please define the abbreviations “GM” and “GN” on first use in the abstract and in the main text. To follow a similar logic of the abbreviation of the terms “medical” and “non-medical” (i.e., N and NM, respectively), consider changing the abbreviation from GN to GNM.

Page 2, line 35 and page 8, line 149. “The median time for single decision was in all categories longer in the GN group (5,28 [4.91-5.66] vs. 4.75 [4.31-5.18] sec, 6.33 [5.96-6.71] vs. 4.10 [3.95-4.26] sec, 5.74 [5.31-6.17] vs. 4.60 [4.27-4.93] sec, 3.01 [2.83-3.20] vs. 3.87 [3.67-4.06] for the E, G, A and I, in the GN and GM groups, respectively).”. Is the median time of 3.01 seconds related to the GM group? If yes, consider placing this number after the median time of 3.87 seconds, as in the previous categories (i.e., E, G and A) the median time of the GN group was mentioned first.

Page 2, line 39. The Cohen’s kappa coefficient for NM mentioned in the text (i.e., 0.376) does not match the coefficient mentioned the Table 2 for the same group (i.e., 0.348). Please clarify.

Page 4, line 54. “Machine Learning (ML) has emerged as an important tool for healthcare, particularly when it comes to training”. Please explain better this sentence.

Page 4, line 73. There is no need to abbreviate “optical coherence tomography” as the authors don’t use the abbreviation elsewhere in the text.

Page 6, line 99 and Table 1. Please explain better when the criteria “Artefacts” is fulfilled. Is the presence of any artefact enough to meet the criteria? Or, as in the EyePACS grading system, the criteria is fulfilled when the image is “sufficiently free of artifacts (such as dust spots, arc defects, and eyelash images) to allow adequate grading”? In the latter case, clinical knowledge would likely be needed to allow judgment of which artefacts are clinically significant to interfere with grading.

Page 6, line 105. “The images were previously labeled for image quality by two experts (KLLF and GMS) using the standards described above.” It would be interesting to know more about the expertise level of the graders KLLF and GMS (e.g., are they ophthalmologists or retina specialists?) and how the disagreements between them were resolved (e.g., arbitration, adjudication). According to Krause et al (2018), when establishing a reference standard, these factors can affect grader variability.

Page 6, line 103. Please explain better when the image is graded as “Insufficient for grading”. If, for a given image, all 4 quality criteria are not met, when will this image be classified as “insufficient” instead of “adequate”?

Page 8, line 166. “There was no consensus among our graders on the importance of the missing optic nerve head”. Please clarify this sentence and why there was a lack of consensus as the optic disc head is one of the key anatomical features in the color fundus photograph and is part of the image field criteria.

Page 9, line 175. Consider rewriting this sentence and using short statements to better express the information.

Page 9, line 182. Consider removing “relatively high number of participants with non-medical background” as only 4 subjects were included in this group.

Page 9, line 189. “In the good, adequate and insufficient groups the grading took longer for the participants without medical background.” Please clarify this sentence as in the results section (page 7, line 150) the authors mention that “the median time for single decision was in all categories longer in the GN group”, including the category “excellent”.

Page 9, line 192. “Our results show that a high level of repeatability can be achieved already by a very short training period and smaller set of retinal photographs, even in the case of medically untrained people.” Repeatability may not be the most appropriate term because it refers to the variation in repeat measurements made by the same instrument or method over a short period of time (Barlett and Frost 2008). This does not seem to be the case of this study, in which the experiment (i.e., grading of multiple images) was performed only once.

Page 10, line 202-215. Consider removing these two paragraphs as they contain information already mentioned in the introduction rather than the context and relevance of the results.

Page 11, line 235. “In our study, it took non-medical graders in median 17 minutes to grade the first 200 images”. Please remove the word “first” as there were only 200 images in the dataset.

Page 11, line 235 and 247. Based on the data presented, it is not possible to conclude that faster grading is a consequence of the learning curve.

Page 11, line 236-238. Remove commas after “estimate” and “ophthalmology”.

Page 11, line 242. “We included only 9 volunteers for the assessment of the labeling task”. Please clarify the number of study participants as in the method section the authors mention (Page 6, line 117) that the images were evaluated by 8 volunteers.

Page 12, line 264-267. “With the increasing role of crowdsourcing in numerous projects harnessing human intelligence, in the context of health-care data the question arises how safe is it to leave diagnosing a medical condition to “untrained” personnel or rather use a machine to perform this task.” Consider rewriting this sentence and using short statements to better express the information.

Page 12, line 270. Please add a conclusion supported by the study results as the last paragraph contains mainly general statements rather than a conclusion about the work itself.

References

Krause, Jonathan, Varun Gulshan, Ehsan Rahimy, Peter Karth, Kasumi Widner, Greg S. Corrado, Lily Peng, and Dale R. Webster. 2018. “Grader Variability and the Importance of Reference Standards for Evaluating Machine Learning Models for Diabetic Retinopathy.” Ophthalmology 125 (8): 1264–72.

Bartlett, J. W., and C. Frost. 2008. “Reliability, Repeatability and Reproducibility: Analysis of Measurement Errors in Continuous Variables.” Ultrasound in Obstetrics & Gynecology: The Official Journal of the International Society of Ultrasound in Obstetrics and Gynecology 31 (4): 466–75.

Reviewer #2: The largest weakness of this study is in the task chosen to assess grader agreement. Image quality is inherently subjective. Even when provided with a grading guideline. Why not instead pick a more objective task to evaluate grader agreement?

Furthermore you do not describe how ground truth was arbitrated or adjudicated between the two specialists, this crucial.

Due to the inherent subjectivity of this task, the author's should have graders select each criteria as y/n (adequate focus, artefacts, etc), then compile and codify to your quality grades. Otherwise, I see this as a poor study design (as the graders likely are not following the guidelines well, letting their subjective "quality" biases dominate, and leading to a poor kappa). Much easier to pick a different task to evaluate agreement.

We're there contrast and brightness adjustments in the grading tool?

How was the ground truth set it arbitration adjudication etc

Line 164 . Please explain what is meant exactly by: Each participant reported positively on the learning experience.

Line 176 please clarify why this is semi automated here and throughout, is there active learning? I'm not sure how this is automated and would avoid this buzzword.

Line 192, please remove repeatability, you did not test this here, you tested agreement

Results: Please emphasize Kappas, they are not reported here.. that's the main outcome, not the times in my opinion.

6. PLOS authors have the option to publish the peer review history of their article (what does this mean?). If published, this will include your full peer review and any attached files.

Reviewer #1: No

Reviewer #2: **Yes: **Edward Korot

---

## [Author Response · Author response to Decision Letter 0]

21 Apr 2022

Rebuttal letter

PONE-D-21-35940

The assessment of fundus image quality labeling reliability among graders with different backgrounds

Journal Requirements

5. We note that Figures 1A, 1B, 1C 1D, 2A and 2B in your submission contain copyrighted images. All PLOS content is published under the Creative Commons Attribution License (CC BY 4.0), which means that the manuscript, images, and Supporting Information files will be freely available online, and any third party is permitted to access, download, copy, distribute, and use these materials in any way, even commercially, with proper attribution. For more information, see our copyright guidelines: http://journals.plos.org/plosone/s/licenses-and-copyright.

Authors’ response: We would like to state here that all images presented in the manuscript were produced by the authors and therefore no permission of the copyright holder is required.

We would like to thank both reviewers for their insightful comments. Please find below our detailed answers for all points raised in the review.

Reviewer #1

Thank you for giving me the opportunity to review the article. I hope the following suggestions will be helpful and will strengthen the work.

The study evaluates the image quality assessment of color fundus photographs among graders with medical and non-medical backgrounds. The topic is interesting and relevant considering the scarcity of labeled data for the development of machine learning models and the importance of imaging quality for the accuracy and applicability of the algorithms. The manuscript is generally well presented, but there are several issues which needs to be addressed:

Authors’ response: We would like to thank Reviewer #1 for his/her comprehensive revision and precise suggestions. We do feel they contributed a substantial improvement of our work. 

Major comments

Page 4, line 51-60. Quotation marks should be used for direct quotes. Consider rewriting the first paragraph of the Introduction as the following sentences were written using the same words as the references:

- Page 4, line 51. “Medical imaging is expanding globally at an unprecedented rate leading to an ever-expanding quantity of data that requires human expertise and judgement to interpret and triage”. De Fauw et al (2018).

- Page 4, line 55. “Benchmark datasets are essential for computational research in healthcare. These datasets should be created by intentional design that is mindful of social and health system priorities. If a deliberate and systematic approach is not followed, not only will the considerable benefits of clinical algorithms fail to be realized, but the potential harms may be regressively incurred across existing gradients of social inequity.” (T. Panch et al. 2020)

Authors’ response: We completely agree with Reviewer #1 and are thankful for raising these points. We found the above quotes from the given references of pivotal importance. Together with other points addressed in both reviews, we restructured the entire first paragraph of the introduction. (See page 3, lines 47-55)

The study was conducted “in an effort to model the democratization of retinal image labeling” (page 5, line 78). To do that, the authors use a phyton grading tool that requires some degree of programming knowledge (page 8, line 162), which could make it difficult for people without specific computing skills to participate in the image labeling process.

Authors’ response: We agree with Reviewer #1 on the controversy of the above statements; therefore, we provided a more detailed explanation of the mentioned problem. 

As for the need for specific computing skills, we stated that all our volunteers had at least intermediate level computer skills. (page 5, line 121) Most of them (6 of 8) had no previous experience with programming in python and the majority found its use easy and intuitive. In our opinion, in case of a real-world scenario most of the participants applying for such a grading task would most probably have at least as good computer (or even minimal programing skills) as our 8 participants had. 

In an effort to provide an even simpler way for grading and to be able to reach a potentially broader circle of volunteers we are currently working on a web-based annotation application, as well.

According to the above, we applied certain changes to the text for clarification. (See page 8, line 181; page 12, line 287)

The imaging grading criteria used in the study take into consideration anatomical structures (e.g., optic nerve head, fovea and macula), and imaging artefacts (e.g., eyelashes and fingerprints) whose identification require a clinical background or specific training and a certain degree of experience. According to the authors, “prior to grading the participants received a tutorial consisting of oral explanation of the task, supported by a pdf document describing the grading system with sample images and the description of the GUI.” (Page 6, line 120). In addition to the tutorial on image quality requirements, has any training been carried out to enable participants to recognize retinal structures and artefacts? If yes, detailed information on how the training was carried out should be provided (e.g., hours of training and grading guidelines). If not, the interpretation of the grading criteria by participants without medical background would be affected, as well as the results of the study.

Authors’ response: We agree with the need for more transparency about grading instructions and training. We provided the participants a written tutorial that included detailed explanation on retinal anatomy as well as the definition of various imaging artefacts. 

One of our main purposes and the motivation of the study were to investigate whether there was a considerable disadvantage in outsourcing the grading task to people without medical background. We are confident that our participants, following the aforementioned tutorial, were able to distinguish between retinal structures such as macula, optic nerve head and retinal blood vessels and to identify different types of artefacts. 

As requested by Reviewer #2 we performed an additional analysis of each single grading criteria to provide a more objective evaluation. 

We applied the above changes to the manuscript (see page 5, line 122) and submitted our tutorial pdf as supplementary material.

Page 7, line 130. Was the agreement calculated between study participants or between each participant and the ground truth (i.e., expert’ labels)? Was the weighted kappa applied to assess agreement when grading image quality (an ordinal outcome)?

Authors’ response: Originally, we calculated Cohen’s Kappa between each participant and the expert labeled data. We agree with the Reviewer that Cohen’s weighted Kappa is more suitable for our study design and, therefore, recalculated our results using weighted Kappa values which are now presented in the revised version of the manuscript (See page 2, line 34; page 6, line 136; page 7, line 165). 

Page 9, line 186. “The time necessary for the first 50 images was higher than for the last 50 images which shows the learning curve of the grading task.” Based on the data presented, it is not possible to conclude that faster grading is a consequence of the learning curve as there are other possible explanations for this result (e.g., grading fatigue).

Authors’ response: We agree with Reviewer #1 and changed the statements (See page 9, line 215; page 11, line 252) regarding the time difference between grading the first and last 50 images. We would, however, like to add that the relatively short processing time (approximately 20 minutes for the original task) and a total number of 200 images would not reach the typical level of exhaustion associated with grader decision fatigue, as shown by Waite et al. 

Waite S, Kolla S, Jeudy J, Legasto A, Macknik SL,Martinez-Conde S, Krupinski EA, Reede DL. “Tired in the Reading Room: The Influence of Fatigue in Radiology”J Am Coll Radiol.2016 https://doi.org/10.1016/j.jacr.2016.10.009

As we later mention, we also agree with Reviewer #1 that a learning curve cannot be observed, therefore, we omitted the aforementioned phrase. (See page 9, line 215; page 11, lines 252 and 269)

Minor comments

Page 2, line 28. “The performance of the grading was timed and the accuracy was assessed. Accuracy was also assessed with the excellent and good categories merged.” Since kappa is not a measure of accuracy, it is not possible to say that the accuracy was assessed.

Authors’ response: We agree with Reviewer #1 on this comment and applied a correction to the text (See page 2, line 30).

Page 2, line 32 and page 7, line 140. Please define the abbreviations “GM” and “GN” on first use in the abstract and in the main text. To follow a similar logic of the abbreviation of the terms “medical” and “non-medical” (i.e., N and NM, respectively), consider changing the abbreviation from GN to GNM.

Authors’ response: We agree with the Reviewer and applied the recommended changes (See page 2, lines 27, 33; page 5, line 119; page 6, line 144; page 7, lines 152, 161-170; page 8, lines 175-178 and 197).

Page 2, line 35 and page 8, line 149. “The median time for single decision was in all categories longer in the GN group (5.28 [4.91-5.66] vs. 4.75 [4.31-5.18] sec, 6.33 [5.96-6.71] vs. 4.10 [3.95-4.26] sec, 5.74 [5.31-6.17] vs. 4.60 [4.27-4.93] sec, 3.01 [2.83-3.20] vs. 3.87 [3.67-4.06] for the E, G, A and I, in the GN and GM groups, respectively).”. Is the median time of 3.01 seconds related to the GM group? If yes, consider placing this number after the median time of 3.87 seconds, as in the previous categories (i.e., E, G and A) the median time of the GN group was mentioned first.

Authors’ response: We thank Reviewer #1 for having this pointed out. We applied the recommended change and rewrote the abstract. (See page 2 and page 7, line 159).

Page 2, line 39. The Cohen’s kappa coefficient for NM mentioned in the text (i.e., 0.376) does not match the coefficient mentioned the Table 2 for the same group (i.e., 0.348). Please clarify.

Authors’ response: We thank Reviewer #1 for this comment, we omitted this statement on page 2 and created a new table (Table 3 on Page 17) with the calculated weighted kappa values for the better transparency of our results.

Page 4, line 54. “Machine Learning (ML) has emerged as an important tool for healthcare, particularly when it comes to training”. Please explain better this sentence.

Authors’ response: We would like to refer to the entirely rewritten first paragraph of the introduction which also contains an explanation of this point. (See page 3, lines 47-55)

Page 4, line 73. There is no need to abbreviate “optical coherence tomography” as the authors don’t use the abbreviation elsewhere in the text.

Authors’ response: We thank Reviewer #1 for pointing this out, we now deleted the abbreviation. (See page 3, line 68)

Page 6, line 99 and Table 1. Please explain better when the criteria “Artefacts” is fulfilled. Is the presence of any artefact enough to meet the criteria? Or, as in the EyePACS grading system, the criteria is fulfilled when the image is “sufficiently free of artifacts (such as dust spots, arc defects, and eyelash images) to allow adequate grading”? In the latter case, clinical knowledge would likely be needed to allow judgment of which artefacts are clinically significant to interfere with grading.

Authors’ response: We extended the explanation in the manuscript (See page 12, line 274; page 14 – Table 1) and would like to refer to our detailed image grading tutorial containing exemplary images. (Suppl. 1) In concordance with our aims for this pilot, we considered the presence of the artefacts enough to meet the criteria regardless of their effect on the image grading possibilities.

Page 6, line 105. “The images were previously labeled for image quality by two experts (KLLF and GMS) using the standards described above.” It would be interesting to know more about the expertise level of the graders KLLF and GMS (e.g., are they ophthalmologists or retina specialists?) and how the disagreements between them were resolved (e.g., arbitration, adjudication). According to Krause et al (2018), when establishing a reference standard, these factors can affect grader variability.

Authors’ response: We thank the Reviewer for this important comment. We added a detailed explanation to our paper, accordingly and inserted the suggested reference. (See page 5, line 102)

Page 6, line 103. Please explain better when the image is graded as “Insufficient for grading”. If, for a given image, all 4 quality criteria are not met, when will this image be classified as “insufficient” instead of “adequate”?

Authors’ response: We would like to thank Reviewer #1 for this important question. We graded images as “insufficient” when a part of the image or the whole image was not adequate for grading retinal lesions at all (and thus all 4 quality criteria were not met). To make this clear, we also included a detailed explanation in our manuscript (see page 5 line 98). This was also explained in our grading tutorial (Suppl. 1):

Insufficient: One or more retinopathy lesions cannot be graded.

In case the third generation branches within one optic disk diameter near the fovea and the optic nerve head cannot be identified, the images should be considered inadequate for grading.

Images lacking visibility of more than 50% of the depicted retinal field should be considered inadequate for grading.

Driven by the above question, we also investigated merging the “adequate” and “insufficient” groups and found that it resulted in a further improvement of the inter-rater agreement (0.665 and 0.670 for the first and second grading round respectively). (See page 7 and 8, lines 165-178 and page 17 – Table 3) 

Page 8, line 166. “There was no consensus among our graders on the importance of the missing optic nerve head”. Please clarify this sentence and why there was a lack of consensus as the optic disc head is one of the key anatomical features in the color fundus photograph and is part of the image field criteria.

Authors’ response: We thank Reviewer #1 for this important remark. The sentence can indeed be misunderstood and refers to our internal debate regarding the importance of the inclusion of the ONH on the images that could, in turn, enable the screening for glaucoma and increase the precision of DR screening. However, our primary goal was the assessment of general image quality and thus the ONH has been graded within the image field definition category as an equally important image quality factor as focus, illumination and artefacts. 

Accordingly, we made our point clearer in the text. (See page 8, line 193; page 11, line 270).

Page 9, line 175. Consider rewriting this sentence and using short statements to better express the information.

Authors’ response: We agree with Reviewer #1 and applied the suggested change (See page 9, line 204).

Page 9, line 182. Consider removing “relatively high number of participants with non-medical background” as only 4 subjects were included in this group.

Authors’ response: We agree with Reviewer #1 and applied the suggested change (See page 9, line 211).

Page 9, line 189. “In the good, adequate and insufficient groups the grading took longer for the participants without medical background.” Please clarify this sentence as in the results section (page 7, line 150) the authors mention that “the median time for single decision was in all categories longer in the GN group”, including the category “excellent”.

Authors’ response: We thank Reviewer #1 to point this out. We applied the suggested clarification. (See page 10, line 223)

Page 9, line 192. “Our results show that a high level of repeatability can be achieved already by a very short training period and smaller set of retinal photographs, even in the case of medically untrained people.” Repeatability may not be the most appropriate term because it refers to the variation in repeat measurements made by the same instrument or method over a short period of time (Barlett and Frost 2008). This does not seem to be the case of this study, in which the experiment (i.e., grading of multiple images) was performed only once.

Authors’ response: We agree with Reviewer #1 on this comment and changed the term repeatability for agreement. We also extended our list of references with the suggested paper. (See page 10, line 226)

Page 10, line 202-215. Consider removing these two paragraphs as they contain information already mentioned in the introduction rather than the context and relevance of the results.

Authors’ response: We agree with Reviewer #1 on this comment and deleted the suggested paragraphs. We also rewrote the entire first paragraph of the introduction in order to avoid repeating the same information. (See page 3, lines 47-55)

Page 11, line 235. “In our study, it took non-medical graders in median 17 minutes to grade the first 200 images”. Please remove the word “first” as there were only 200 images in the dataset.

Authors’ response: We agree with Reviewer #1 and applied the suggested change (See page 11, line 252).

Page 11, line 235 and 247. Based on the data presented, it is not possible to conclude that faster grading is a consequence of the learning curve.

Authors’ response: We agree with Reviewer #1 that a learning curve cannot be observed, therefore we omitted the mentioned sentences. (See page 11, lines 252 and 269)

Page 11, line 236-238. Remove commas after “estimate” and “ophthalmology”.

Authors’ response: We thank Reviewer #1 for pointing this out. We applied the suggested correction. (See page 11, lines 252 and 254)

Page 11, line 242. “We included only 9 volunteers for the assessment of the labeling task”. Please clarify the number of study participants as in the method section the authors mention (Page 6, line 117) that the images were evaluated by 8 volunteers.

Authors’ response: We thank Reviewer #1 for pointing out this typo that we corrected in the text. (See page 11, line 263)

Page 12, line 264-267. “With the increasing role of crowdsourcing in numerous projects harnessing human intelligence, in the context of health-care data the question arises how safe is it to leave diagnosing a medical condition to “untrained” personnel or rather use a machine to perform this task.” Consider rewriting this sentence and using short statements to better express the information.

Authors’ response: We agree with Reviewer #1 and simplified the sentence by using short statements. (See page 12, line 294)

Page 12, line 270. Please add a conclusion supported by the study results as the last paragraph contains mainly general statements rather than a conclusion about the work itself.

Authors’ response: We thank the Reviewer for this valuable remark and agree that a more effective paragraph with the conclusion of our work is necessary. In line with this we rewrote the last paragraph (See page 12, line 298).

References

Krause, Jonathan, Varun Gulshan, Ehsan Rahimy, Peter Karth, Kasumi Widner, Greg S. Corrado, Lily Peng, and Dale R. Webster. 2018. “Grader Variability and the Importance of Reference Standards for Evaluating Machine Learning Models for Diabetic Retinopathy.” Ophthalmology 125 (8): 1264–72.

Bartlett, J. W., and C. Frost. 2008. “Reliability, Repeatability and Reproducibility: Analysis of Measurement Errors in Continuous Variables.” Ultrasound in Obstetrics & Gynecology: The Official Journal of the International Society of Ultrasound in Obstetrics and Gynecology 31 (4): 466–75.

Authors’ response: We included these references in the manuscript in the above mentioned sections.

Reviewer #2

Authors’ response: We would also like to thank Reviewer #2 for the important points raised and for his efforts to improve our work.

Reviewer #2: The largest weakness of this study is in the task chosen to assess grader agreement. Image quality is inherently subjective. Even when provided with a grading guideline. Why not instead pick a more objective task to evaluate grader agreement?

Due to the inherent subjectivity of this task, the author's should have graders select each criteria as y/n (adequate focus, artefacts, etc), then compile and codify to your quality grades. Otherwise, I see this as a poor study design (as the graders likely are not following the guidelines well, letting their subjective "quality" biases dominate, and leading to a poor kappa). Much easier to pick a different task to evaluate agreement.

Authors’ response: We completely agree with Reviewer #1 regarding his concern about the inherent subjectivity of our study. For this reason, we conducted a second grading round with the same subjects based on the suggestion of the Reviewer. We adapted our grading guideline (Suppl. 1) and asked the participants to select each grading criterion for the four image quality groups. Thus, the graders could choose between 14 single criteria (e.g. “Focus optimal”, “Focus unsharp”, “Illumination optimal”, “Illumination too dark”, “Illumination too light/overexposed” etc.).

The details of this extra round of assessment are described in the Methods and Results section, accordingly (See page 6, lines 126 and 134; Table 2 – page 16). 

According to our results (page 2, line 34; page 7, line 165; page 17 – Table 3) the above, more objective classification helped our graders reach higher agreement: median weighted kappa 0.564 vs. 0.594 (4 groups) and 0.637 vs. 0.667 (3 groups). 

However, it is important to point out two facts regarding our work. First, our study was merely a pilot in nature that should allow a basis for future assessments. Second, the ultimate benefit of our work should be the robust identification of images with poor quality in a screening setting. Based on our results, even with the inherent subjectivity taken into consideration, providing 3 categories for fundus image quality grading could be sufficient, simple and fast without the need of multiple predefined labels. We believe that images that are of poor quality and thus are certainly not sufficient for screening for retinal diseases can be reliably found and be eliminated that could, in turn, enable a better training for future AI algorithms.

We're there contrast and brightness adjustments in the grading tool?

Authors’ response: We thank the Reviewer for this important question. There were no zooming, brightness or contrast changing options available. Currently, we are working on a web-based application which would contain additional zoom and brightness adjustments. We also stated it in the revised manuscript. (See page 12, line 287)

Furthermore you do not describe how ground truth was arbitrated or adjudicated between the two specialists, this crucial.

How was the ground truth set it arbitration adjudication etc

Authors’ response: We would like to thank Reviewer #2 for this comment. A detailed explanation was added to the manuscript (page 5, line 102): “The images were previously labeled for image quality by two experts and served as the reference for the analysis (KLLF – board certified ophthalmologist with experience in retinal imaging – and GMS – board certified senior retina specialist with retinal image grading experience in the telemedical screening of diabetic retinopathy) using the standards described above. In case of any disagreement, GMS’s decision served as reference.”

Line 164 . Please explain what is meant exactly by: Each participant reported positively on the learning experience.

Authors’ response: We added a detailed description of the feedback received from the participants. (See page 8, line 189)

Line 176 please clarify why this is semi automated here and throughout, is there active learning? I'm not sure how this is automated and would avoid this buzzword.

Authors’ response: We agree with Reviewer #2 and deleted the term “semi-automated” in order to avoid such a misunderstanding. (See page 9, line 204).

Line 192, please remove repeatability, you did not test this here, you tested agreement

Authors’ response: We agree with Reviewer #2 on this comment and applied a correction. (See page 10, line 226)

Results: Please emphasize Kappas, they are not reported here.. that's the main outcome, not the times in my opinion.

Authors’ response: We included a more detailed elaboration of the Kappas, including the assessment of weighted Cohen’s Kappa values. The changes can be found on page 2, line 34 and page 7, line 165.

---

## [Decision Letter · Decision Letter 1]

27 Jun 2022

The assessment of fundus image quality labeling reliability among graders with different backgrounds

PONE-D-21-35940R1

Dear Dr. Somfai,

We’re pleased to inform you that your manuscript has been judged scientifically suitable for publication and will be formally accepted for publication once it meets all outstanding technical requirements.

Kind regards,

Prof. Andrzej Grzybowski

Academic Editor

PLOS ONE

Additional Editor Comments (optional):

Thank you for submitting your valuable paper to our journal.

**Comments to the Author**

1. If the authors have adequately addressed your comments raised in a previous round of review and you feel that this manuscript is now acceptable for publication, you may indicate that here to bypass the “Comments to the Author” section, enter your conflict of interest statement in the “Confidential to Editor” section, and submit your "Accept" recommendation.

Reviewer #1: All comments have been addressed

Reviewer #2: All comments have been addressed

2. Is the manuscript technically sound, and do the data support the conclusions?

Reviewer #1: Yes

Reviewer #2: Yes

3. Has the statistical analysis been performed appropriately and rigorously? 

Reviewer #1: Yes

Reviewer #2: Yes

4. Have the authors made all data underlying the findings in their manuscript fully available?

Reviewer #1: No

Reviewer #2: Yes

5. Is the manuscript presented in an intelligible fashion and written in standard English?

Reviewer #1: Yes

Reviewer #2: Yes

6. Review Comments to the Author

Reviewer #1: I appreciate the authors’ effort to address the suggestions. The adjustments (e.g., changes in statistical analysis and more details about the grading protocol) significantly improved the manuscript.

Just one minor comment:

Page 9, line 209. Please replace “repeatability” with “agreement” as “repeatability” was not tested in this study.

Reviewer #2: Thanks for addressing all comments, the manuscript is now much improved. I appreciate the thoroughness of additional analyses.

7. PLOS authors have the option to publish the peer review history of their article (what does this mean?). If published, this will include your full peer review and any attached files.

Reviewer #1: No

Reviewer #2: **Yes: **Edward Korot

---

## [Editor Report · Acceptance letter]

13 Jul 2022

PONE-D-21-35940R1 

The assessment of fundus image quality labeling reliability among graders with different backgrounds 

Dear Dr. Somfai:

I'm pleased to inform you that your manuscript has been deemed suitable for publication in PLOS ONE. Congratulations! Your manuscript is now with our production department. 

Kind regards, 

on behalf of

Dr. Andrzej Grzybowski 

Academic Editor

PLOS ONE